# A Study on the Development of Carbon Fiber with Electromagnetic Wave Shielding Performance and Sizing Removal State Measurement Algorithm Using Image Processing

Joon-Ho Cho

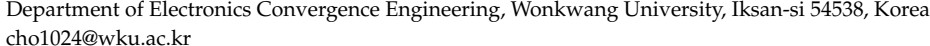

Department of Electronics Convergence Engineering, Wonkwang University, Iksan-si 54538, Korea; cho1024@wku.ac.kr

**Abstract:** In this paper, nickel-plated carbon fiber was fabricated by a dry process method to improve electromagnetic wave shielding performance. In general, carbon fiber is wrapped in a polymer type in manufacturing and is used after removing the sizing in the pretreatment step for dry coating. The existing sizing removal method was used by removing only with a solution or only with compressed air. In this paper, the method of solution and compressed air (hybrid) was added. The state in which the sizing was removed was determined only by the know-how of the experienced person, and in this paper, it is proposed to represent the numerical value by applying the image processing surface analysis technique. As a result, it was possible to numerically indicate that the hybrid method was excellent among the sizing removal methods and it was possible to manufacture the nickel-plated carbon fibers (30 μm, 40 μm, and 100 μm) by a roll-to-roll sputtering method on the sizing-removed spreading carbon fiber roll. The electromagnetic wave shielding performance of 100 nickel-coated carbon fiber measured by the Korea Testing Laboratory showed the highest electromagnetic wave shielding performance from 66.7 (dB) to 73.2 (dB). This is similar to the electromagnetic wave shielding rate of copper, so it can be used as a cable for EV/HEV vehicles, and it is expected to have a great effect of improving the bending characteristics and disconnection phenomenon and improving the lifespan compared to the existing copper wire.

**Keywords:** carbon fibers; electromagnetic; image processing; relationship coefficient; shielding



## 1. Introduction

As the problems of depletion of energy stored in the earth and air pollution of the global environment have emerged, a period of major change is coming to the automobile market. Vehicles that use non-fossil fuels (EVs/HEVs) rather than vehicles that use conventional fossil fuels are attracting attention in the global market as the next-generation ground transportation means [1,2].

One of the most important materials in composite materials is carbon fiber. High-performance carbon fiber has very high tensile strength and modulus, and has excellent abrasion resistance, lubricity, and conductivity.

Electric wires used in EV/HEV vehicles account for about 5 to 10 wt% of the total weight of the vehicle, and in order to reduce the weight and prevent malfunction of the vehicle control circuit, it must possess electromagnetic wave shielding performance. Development of a shielding shield using carbon fiber and development of a shielded cable by replacement can reduce the weight of the shielded cable itself, and in the case of a shielded shield, it can retain high corrosion resistance [3–15].

In order to apply the carbon fiber shielding cable to automobiles, the current carbon fiber resistivity of about $1 \times 10^{-3}$ (Ω·cm) needs to be lowered to about $2 \times 10^{-4}$ (Ω·cm). It is necessary to develop technology for electromagnetic wave shielding tape applied

with carbon fiber and spread method using it and shield braiding applied with braiding process. Carbon fiber surface treatment technology, carbon fiber metal coating technology, surface sizing technology, carbon fiber spreading and slitting technology, carbon fiber elements such as electromagnetic shield manufacturing technology by braiding technology, and carbon fiber electromagnetic shielding cable evaluation technology should be comprehensively studied.

In this paper, nickel-plated carbon fibers were produced using an algorithm that numerically indicates the state of removal of sizing through image analysis of carbon fibers during this process and a roll-to-roll sputtering method is presented.

## 2. Metal Coating Technology and Sizing Removal Image Processing of Carbon Fiber for Electromagnetic Wave Shielding Cable

Carbon fiber has been widely used as an adsorbent or a filler material for improving properties of polymer materials due to its high strength and conductivity. When the surface of carbon fiber with these properties is improved, the bonding strength between the polymer matrix and the carbon fiber is increased, so that the physical properties of the composite material are improved, or the adsorption property is improved, and further improved effects can appear. In order to improve the electrical conductivity of carbon fibers, a nickel plating method is being developed on the surface of carbon fibers using electrolytic plating and electroless plating methods. In this paper, the development of dry-type metal-coated carbon fiber for electromagnetic shielding, analysis of sizing removal status using image processing, and electromagnetic shielding performance are dealt with.

### 2.1. Analysis of Pretreatment State of Carbon Fiber Using Image Processing

For dry metal coating on carbon fiber, the carbon on the carbon fiber surface must be exposed. However, in the manufacture of carbon fiber, it comes wrapped in a type of polymer called a sizing agent. This prevents the carbon fibers from falling off and allows them to be used in bundles, but the use of individual carbon fibers requires removal of sizing. To obtain a uniform, high-quality metal-coated surface, the sizing attachment surface should be irradiated and removed by pretreatment. In this paper, the carbon fiber sizing dissolution removal test using toluene, benzene, and cyclohexane; the carbon fiber sizing dissolution removal test using compressed air; and the compressed air injection nozzle method after removal through dissolution, that is, the hybrid method, are used for sizing of carbon fiber removed from the surface. Although the removal status of the removed sizing was expressed in approximate numerical terms, the logical basis was insufficient. In this paper, this insufficient logic is proposed through image processing.

The algorithm proposed in this paper is shown in Figure 1. The first step is to secure a raw material for carbon fiber surrounded by sizing. The second step is to remove the sizing, and there are solution, nozzle pressure, and hybrid methods. The last step is to numerically represent the state of removal of sizing using image processing. The image processing algorithm secures an SEM (scanning electron microscope) image of the carbon fiber from which the sizing has been removed and divides only the carbon fiber part. Here, in the SEM image, the remaining part except for the carbon fiber part can be easily divided into black. In order to compare the removed state of the sizing, carbon fiber parts of the same size were selected, and when the correlation to the texture of the surface was calculated, the removed state of the sizing was numerically represented.

Figure 2 is an image of the state in which the sizing has been removed using dissolution. So far, the sizing removal rate in Figure 2 is expressed as approximately 70%.

Figure 3 shows an image in which the sizing has been removed with compressed air whose shape is changed with a flat nozzle and the spray angle is adjusted. As shown in Figure 3, there are parts where the sizing has not been removed. In the existing method, it is expressed that about 80% of the sizing removal state of the image in Figure 2 has been numerically removed, but it cannot be logically explained.

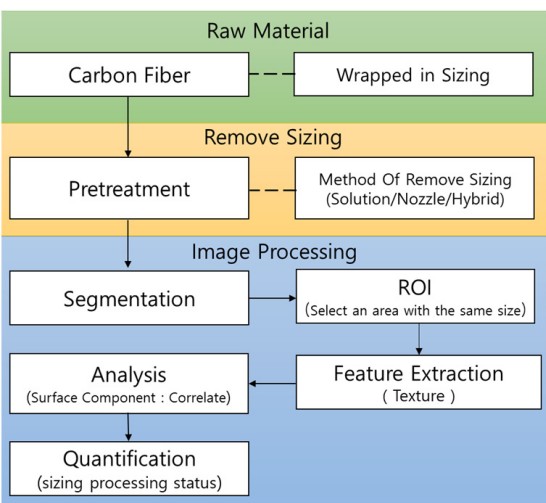

**Figure 1.** Proposed algorithm.

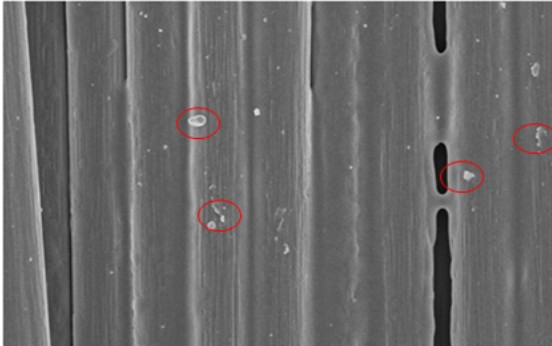

**Figure 2.** Sizing state of carbon fibers.

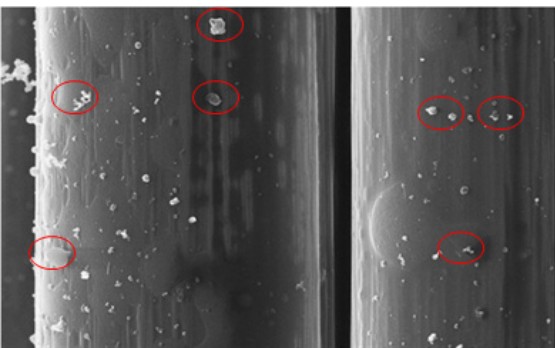

**Figure 3.** Image of sizing removed state by high pressure.

Figure 4 is an image with sizing removed using the hybrid method. It can be seen from the image state that a lot of sizing has been removed compared to Figures 2 and 3. As a result of inquiring with the Carbon Technology Institute, the state in which the sizing was removed was approximately 93%. The percentage of sizing removal status in Figures 2 and 3 is not logical as it is a value based on the know-how of practitioners. In this paper, to provide a logical basis, a texture analysis of the surface was performed using an image processing technique. The proposed method first assumes that the texture analysis of the image surface from which a lot of sizing has been removed is 100, as shown in Figure 4, and represents the image without sizing as a relative numerical value.

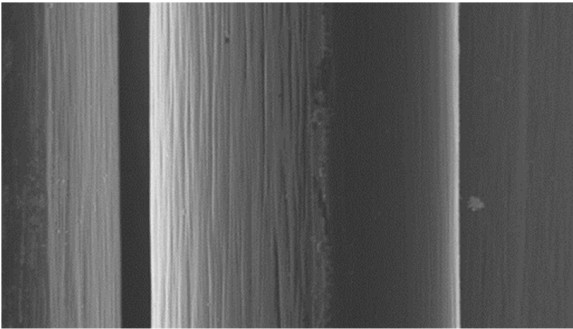

**Figure 4.** Image of a lot of sizing removed state by the hybrid method.

Among the texture analysis methods of image processing, the correlation of Equation (1) is close to 1 for periodic patterns. Therefore, by obtaining the value of Equation (1) from the image of carbon fiber, anyone can numerically represent the state of removal of sizing [16–20].

$$\sum_{i=1}^{k}\sum_{j=1}^{k}\frac{(i-m_r)(i-m_c)p_{ij}}{\sigma_r\sigma_c} \tag{1}$$

where $m_r = \sum_{i=1}^{k} ip(i)$ and $m_c = \sum_{j=1}^{k} jp(j)$ mean the mean calculated along the row and column, respectively. $\sigma_r = \sum_{i=1}^{k}(1-m_r)^2 p(i)$, $\sigma_c \sum_{j=1}^{k}(1-m_c)^2 p(j)$ means the standard deviation calculated along the row and column, respectively.

### 2.2. Spreading Carbon Fiber Roll Manufacturing

Carbon fiber roll manufacturing was carried out under the most excellent conditions, and when the sizing removal process was passed and the spreading device stage was passed, a spreading carbon fiber roll was obtained. 12K carbon fiber was used, and the width was set to 25 mm.

Figure 5 shows that the carbon fiber from which the sizing was removed and was compressed with a metal plate to produce a thickness of 40 mm and a width of 25 mm.

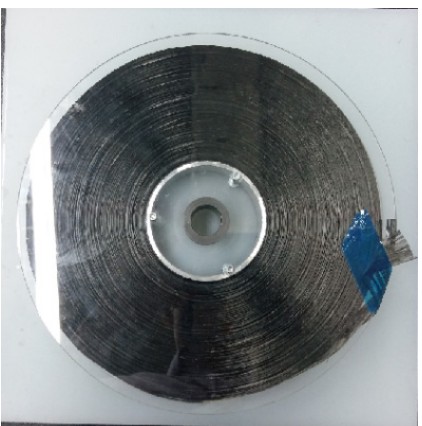

**Figure 5.** Spreading roll of carbon fibers of sizing removed.

### 2.3. Carbon Fiber Coated with Nickel Metal by Dry Coating Method for Electromagnetic Wave Shielding

The production of nickel-coated carbon fiber is carried out by roll-to-roll sputtering and a carbon fiber roll-to-roll feeder. Figure 6 is a roll-to-roll sputter designed and manufactured to enable the maximum feed rate of carbon fiber of 10 M/min and the minimum speed of 0.1 M/min. The transfer was made directly so that it could be transferred in a tight pulling state by adjusting the RPM of the torque motor and adjusting the tension of the brake.

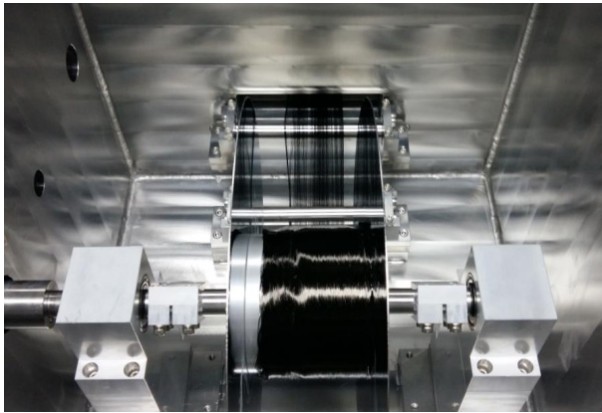

**Figure 6.** Roll-to-roll transfer device.

The order of dry coating on the carbon fiber consists of the step of controlling the progress direction of the carbon fiber and the sputtering operation sequence. The fiber movement direction is adjusted in the order of aligning the rotational direction of the bobbin central axis, adjusting the axial direction of the tension control rotating body, and adjusting the height of the guide rotating body. Figure 7 shows the operation of the carbon fiber rotating roll axial adjustment process.

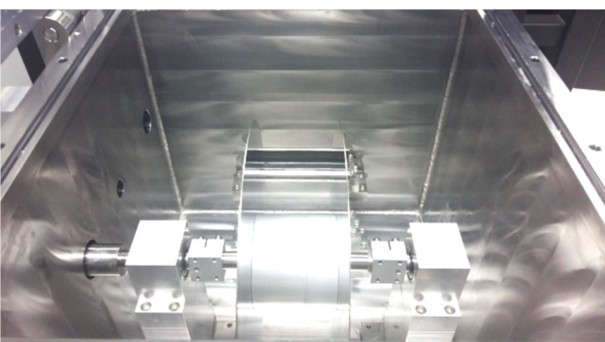

**Figure 7.** Rotating roll sort operation of carbon fiber.

After adjusting the moving direction of the carbon fiber, gas is injected and power is supplied to coat the carbon in a dry manner. The vacuum pressure used here started argon injection at $7 \times 10^{-5}$ torr, purged for 10 min, and generated plasma in the $9 \times 10^{-3}$ torr band. The voltage was 800 VDC and the power was 2.8 kW. Figure 8 is an SEM picture of the coated carbon fiber with the speed set to 10 cm/min.

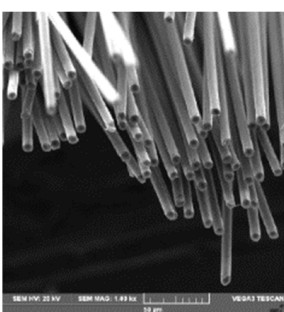
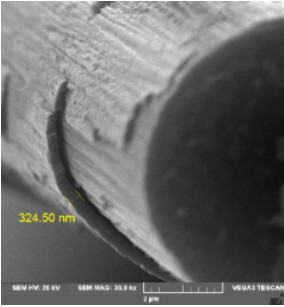

**Figure 8.** Metal-coated carbon fiber SEM measurement.

The electromagnetic shielding (*SE*) test method of the nickel-coated carbon fiber produced in this way was measured according to ASTM D4935 [21,22]. The electromagnetic wave shielding effect of a material is defined in Equation (2) as the ratio of the received

power $p1$ without the shielding material to the received power $p2$ without the shielding material.

$$S.E(dB) = 10log\frac{p1}{p2} \tag{2}$$

## 3. Evaluation Results and Consideration of Electromagnetic Wave Shielding Performance

### 3.1. Analysis Result of Pretreatment State of Carbon Fiber Using Image Processing

In this paper, the texture of the image surface was numerically expressed using the Matlab 2021a version and using the Image tool box. In the image from which the sizing of carbon fiber was removed, the correlation was close to 1, and in the image in which the sizing was not removed, the correlation was close to the value of 0. In the simulation, Figures 2–4 described above were used.

First, Figures 2–4 were separated into values of the same size under the same conditions. Then, using the ROI of digital image processing, it was saved as shown in Figure 9. The correlation of Equation (1) was obtained through image processing of the stored image.

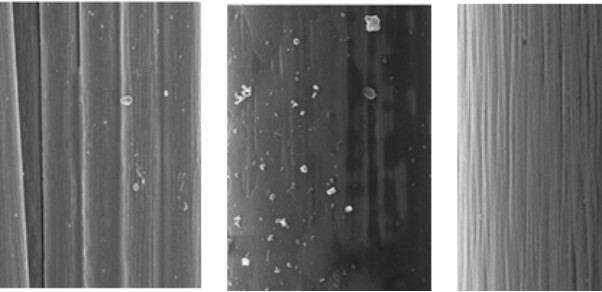

**Figure 9.** Select the ROI range only saved image.

The regular pattern of carbon fiber theoretically approaches 1 as the sizing is removed. As seen in Table 1, sizing removal with solution was 80%, sizing removal with compressed air was 93%, and sizing removed with solution and compressed air was confirmed to be numerically expressed as 96%.

**Table 1.** Sizing delete state of carbon fiber.

|  | Sizing Removal by Solution (Figure 2) | Sizing Removal by Compressed Air (Figure 3) | Sizing Removal by Hybrid Method (Figure 4) |
|---|---|---|---|
| Mean | 107.9765 | 103.5886 | 104.0306 |
| Standard Deviation | 22.3881 | 27.0424 | 40.7007 |
| Relationship coefficient | 0.8005 | 0.9343 | 0.9639 |

### 3.2. Evaluation and Analysis of Electromagnetic Wave Shielding of Carbon Fiber Coated with Nickel Metal Using Dry Process

Formation of the carbon fiber electromagnetic wave shielding layer was evaluated by comparing the method of forming 30 μm and 40 μm thickness. The equipment used to measure the electromagnetic shielding was Network Analyzer (B5071B), Far Field Test Fixture (B-01-N), and Attenuator (1001080447), and Equation (2) was applied for the electromagnetic shielding performance.

Figure 10 is a photograph of a 40-μm metal-coated carbon fiber electromagnetic shielding film used in the test.

Table 2 compares five performance parameters of the existing nickel-plated carbon fiber and the manufactured nickel-plated carbon fiber.

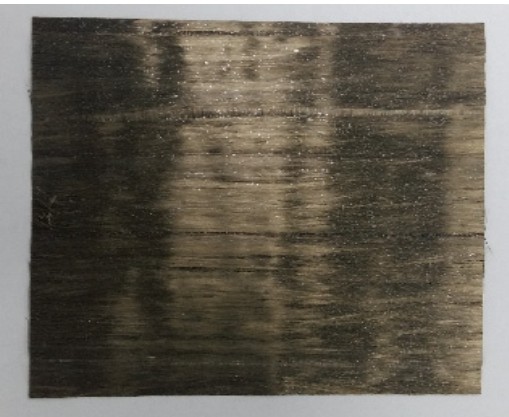

**Figure 10.** Metal-coated carbon fiber electromagnetic wave shielding film.

**Table 2.** Performance of manufactured products.

|  | Unit | Existing Product | Manufactured Product |
|---|---|---|---|
| Volumetric resistance | $\Omega{\cdot}cm$ | $5 \times 10^{-4}$ | $2.72 \times 10^{-5}$ |
| Electromagnetic Shielding rate | dB | 30 | 66.7 |
| Carbon fiber density | $g/cm^3$ | 3.3 | 2.42 |
| Coating thickness | mm | 0.5 | $2 \times 10^{-4}$ |
| Tensile strength | MPa | 5500 | 6400 |

Figure 11 shows the test results of electromagnetic wave shielding by the method of Equation (2), showing that the carbon fiber coated with a thickness of 40 μm has a higher electromagnetic wave shielding rate than the carbon fiber coated with a thickness of 30 μm. The analysis result can be viewed as a phenomenon that occurs because the carbon fiber continues to form the electromagnetic wave transmission portion through the void in the same form.

Figure 12 shows the test results of the electromagnetic wave shielding rate of carbon fiber coated with nickel to a thickness of 100 μm. The test showed a high shielding rate of 66 dB.

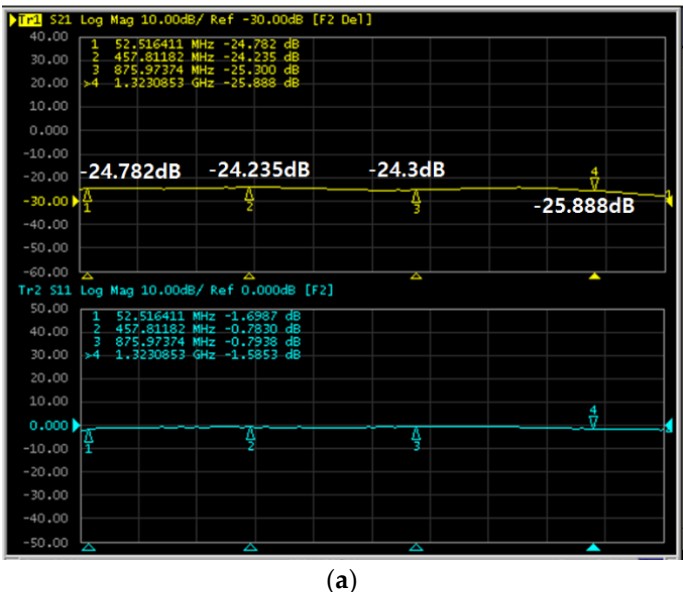

(**a**)

**Figure 11.** *Cont.*

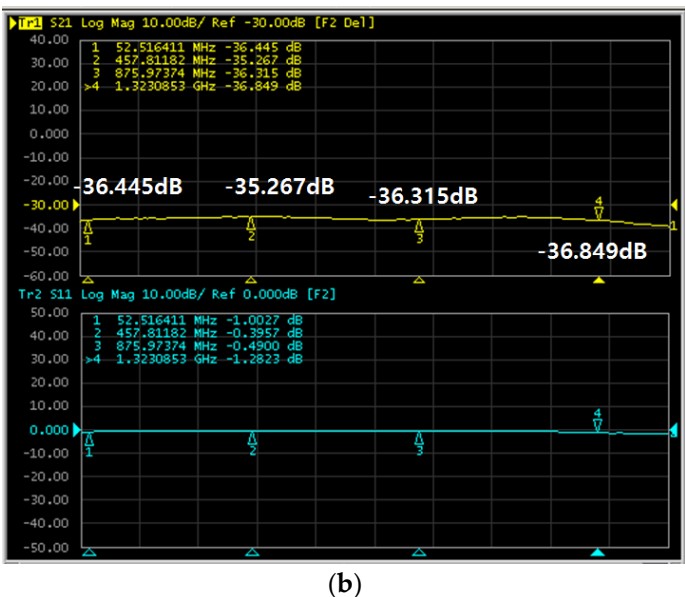

(**b**)

**Figure 11.** (**a**) 30 μm metal-coated carbon fiber; (**b**) 40 μm metal-coated carbon fiber.

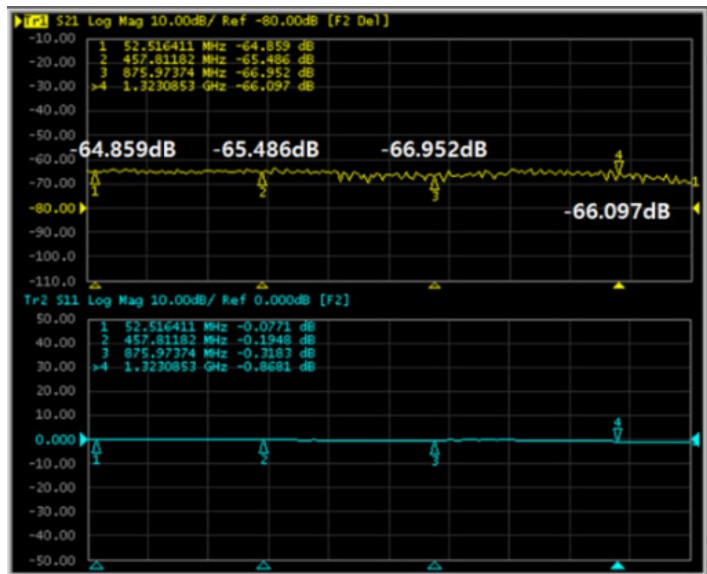

**Figure 12.** Electromagnetic shielding performance of the metal coating carbon fiber 100 μm.

This is similar to thin film foil, and it is also a performance that can be used as a wiring harness for automobiles or an electromagnetic wave blocking film material for military use.

Table 3 shows the measurement results for the measurement conditions of the nickel-coated carbon fiber. The measurement conditions showed an electromagnetic wave shielding effect of 66.96 (dB) in a nickel-coated carbon fiber with a thickness of 100 μm against a plane wave in the frequency range from 30 MHz to 1.5 GHz. As a result of evaluating the electromagnetic wave shielding rate test of nickel-coated carbon fiber with 100 μm thickness that was directly tested in the previous test under the same conditions by the Korea Testing and Research Institute, which is a KOLAS certification body, the lowest 66.7 (dB) and the highest 73.2 (dB) were shown. In terms of electromagnetic shielding effect, 60–90 dB is excellent, and 90–120 dB is judged to be the highest level. Therefore, the manufactured 100 μm thick nickel-coated carbon fiber is a performance that can be used in EV/HEV automobiles.

**Table 3.** Comparison of electromagnetic shielding performance.

| | Measurement Condition (30 MHz–1.5 GHz) | | | |
| | Lowest Electromagnetic Shielding Effect | | Best Electromagnetic Shielding Effect | |
| | Frequency (MHz) | Measures (dB) | Frequency (MHz) | Measures (dB) |
| --- | --- | --- | --- | --- |
| 30 μm | 457.8 | 24.235 | 1323 | 25.888 |
| 40 μm | 457.8 | 35.267 | 1323 | 36.849 |
| 50 μm | 52.5 | 64.859 | 857.97 | 66.952 |

## 4. Conclusions

In this paper, it was confirmed that the sizing removal state can be numerically expressed using the image processing algorithm. As a result of applying the proposed algorithm, the removal of sizing using dissolution was 80%, the removal of sizing using compressed air was 93%, and the hybrid method combining the two was the best at 96%. In the process of manufacturing nickel-coated carbon fiber in a dry method, care must be taken because the nickel coating layer becomes too thick if there is an oxidation problem or insufficient deposition density of the nickel deposition film.

Electromagnetic wave shielding performance was investigated by manufacturing nickel-coated carbon fiber in a dry method. As a result of the analysis of the electromagnetic wave shielding performance, the carbon fiber coated with a thickness of 30 μm showed an electromagnetic wave shielding performance of 24 (dB), the carbon fiber coated with a thickness of 40 μm was 36 (dB), and the carbon fiber coated with a thickness of 100 μm showed an electromagnetic wave shielding performance of 65 (dB). As a result, it was confirmed that the larger the nickel-coated area, the better the electromagnetic wave shielding performance. In particular, carbon fiber coated with a thickness of 100 μm is similar to the shielding rate of copper, so it can be used as an EV/HEV car cable in the future.

**Funding:** This paper was supported by Wonkwnag University in 2021.

**Conflicts of Interest:** The author declares no conflict of interest.

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
