# Peer review of "A Study on the Development of Carbon Fiber with Electromagnetic Wave Shielding Performance and Sizing Removal State Measurement Algorithm Using Image Processing"

_electronics, doi:10.3390/electronics10243128_

Round 1

Reviewer 1 Report

Overall feedback:The work seems to be a good scale but it looks to be related to composite materials research but it looks like partial analysis is done by using image processing but there is no much details about why image processing is needed, no explanation about algorithm and findings by image processing, how it was useful in the study. Need the following clarifications.

  1. In the paper there is no appropriate explanation about the image processing algorithm is being found.
  2. The first step is removing the sizing, I wonder how is the image size is selected, the method to resize the image and the cause to do so.
  3. Second step, acquring specific part-How is the specific part is acquired and why. 
  4. It is said surface component is analyzed using image processing. To analyze the surface, what is the method of image processing applied and why.
  5. How is the sizing percentage is defined for figure 1 & 2.
  6. There are no calculations identified to mean and standard deviation and how is it correlation is close to 1 is defined.
  7. What is the software that is used to define the values as shown in figure 10, 11. Need more clarification with respect to the findings.
  8. How is the electromagnetic sheiding performance values are obtained. Need to provid clarity.

Author Response

First of all, I would like to thank you for reviewing insufficient thesis carefully and pointing out the wrong part. The parts pointed out by your judges were revised and supplemented as follows and reflected 100% in the original thesis. Thank you.

1. In the paper there is no appropriate explanation about the image processing algorithm is being found.

- The proposed algorithm of this paper was added, and related contents were written in lines 84 to 93.

2. The first step is removing the sizing, I wonder how is the image size is selected, the method to resize the image and the cause to do so.

- Images can be divided into objects and backgrounds. Here, the object is the part to be interpreted, and the rest can be called the background. In this paper, the object is made of carbon fiber, and the background is defined as the rest except for the carbon fiber. When interpreting an image, we start by separating objects and backgrounds. In this paper, the first step is to select only objects. This part is called segmentation in image processing. The related contents were written on lines 84 to 93.

3. Second step, acquring specific part-How is the specific part is acquired and why.

- A numerical comparison of the sizing removal condition is to compare carbon fibers of the same size. In this paper, it corresponds to the ROI (Region of Interesting) stage, and related contents were written in lines 84 to 93.

4. It is said surface component is analyzed using image processing. To analyze the surface, what is the method of image processing applied and why.

- Surface analysis in image processing is interpreted as texture, and related equations are expressed as correlations as in equation (1) in the paper.

5. How is the sizing percentage is defined for figure 1 & 2.

- The percentage of sizing removal status in Figures 1 and 2 was roughly defined by the practitioner's know-how. As a result, different values can be expressed by practitioners. In this paper, it is proposed to approach logically by applying image processing techniques.

6. There are no calculations identified to mean and standard deviation and how is it correlation is close to 1 is defined.

- The correlation value can be obtained based on the mean and standard deviation, and in this paper, it is written in addition to lines 127 to 129.

7. What is the software that is used to define the values as shown in figure 10, 11. Need more clarification with respect to the findings.

- The measurement of electromagnetic wave shielding performance is evaluated by the certification body with measuring equipment. In this paper, information related to the measured equipment was added to lines 213 to 216.

8. How is the electromagnetic sheiding performance values are obtained. Need to provid clarity.

- Electromagnetic shielding performance is measured based on Reference No. 21 (ASTM D4935). In this paper, equation (2) corresponds to line 178.

Reviewer 2 Report

The manuscript presents a numeric method to size removal condition of pretreatment of composite material through applying image processing algorithm and nickel - plated carbon fiber fabricated by dry process method to enhance electromagnetic shielding performance.
Results obtained confirmed that the sizing removal state can be numerically expressed using the image processing algorithm. It allows it to be as an vehicles that use non-fossil fuels car cable in the future.

I find the topic interesting and being worth of investigation and the document is well strucutred, organized, fluidly written, has enough background information, the methodology followed is properly explained and is correct, the results are clearly presented and support the conclusions.
Although I propose the following suggestions:
- Abstract requires structuring such as: problem, motivation, aim, methodology, main results, further impact of those results.
- Fig. 10 and 11 should be made bigger.  
- Limitations found in conducting this study should be disclosed.

Author Response

First of all, I would like to thank you for reviewing insufficient thesis carefully and pointing out the wrong part. The parts pointed out by your judges were revised and supplemented as follows and reflected 100% in the original thesis. Thank you.

1. Abstract requires structuring such as: problem, motivation, aim, methodology, main results, further impact of those results.

- The abstract part has been rewritten by reflecting the opinions of the judges.

2. Fig. 10 and 11 should be made bigger.

- The images in Figures 10 and 11 have been greatly edited.

3. Limitations found in conducting this study should be disclosed.

- The most difficult part while conducting this study was reflected in lines 264 to 267.

Round 2

Reviewer 1 Report

There are revisions identified in the paper. There is still needed to clarify about the image processing algorithm in detail, as the major contribution is claimed on stating results obtained using image processing.

Can you please brief in detail about the image processing algorithm mentioned such as

  1. Do you mean Quantification or Quantization? What is this quantification method in your work and how it is done?
  2. Can you breif the result of features extracted and how analysis is concluded? 
  3.  There are so many approached so finding ROI? Which one is being applied in your algorithm?
  4. In the previous review comments, Q6 is not still addressed, I can see the Formulae but there are no calculated values?
  5. "In this paper, this insufficient logic is proposed through image processing", can you please explain why is it not possible with respect to your research?
  6. In the previous review comments, Q5 is not properly answered. It seems most of the image processing work is done manually using expertise than any method applied to identify the finding using image processing techniques?Justify?

Author Response

The completion of our thesis seems to have improved due to the faithful review of the judges. The points pointed out by the judges were revised and supplemented as follows and reflected 100% in the original paper. Thank you.

1.Do you mean Quantification or Quantization? What is this quantification method in your work and how it is done?

-Previously, practitioners expressed the sizing removal state, which is the pre-processing stage of carbon fiber, in approximate numerical values, but the proposed method is numerically expressed by applying the texture processing technique of image processing.

2.Can you breif the result of features extracted and how analysis is concluded?

-Feature extraction refers to extracting an image of the same size for image processing analysis, and the analysis result means analyzing the extracted feature image from a correlation point of view.

3.There are so many approached so finding ROI? Which one is being applied in your algorithm?

-The method applied in this paper proceeded in the order of carbon fiber image-> Segmentation-> ROI-> feature extraction-> Analysis. ROI is applied to interpret images under the same size condition.

4.In the previous review comments, Q6 is not still addressed, I can see the Formulae but there are no calculated values?

-It is presented in Table 1 of this paper.

5."In this paper, this insufficient logic is proposed through image processing", can you please explain why is it not possible with respect to your research?

-In the past, the sizing removal status was interpreted differently depending on the practitioner, but it has the advantage of being able to interpret the sizing removal status based on numerical evidence through image processing.

6.In the previous review comments, Q5 is not properly answered. It seems most of the image processing work is done manually using expertise than any method applied to identify the finding using image processing techniques?Justify?

-The method of expressing the sizing removal status of carbon fiber as a percentage defines the sizing removal status as a percentage of 100%. Then, the practitioner subjectively decided that Figure 2 was 70% and Figure 3 was 80%. So, in this paper, we quantify the sizing removal status as a correlation concept through image processing.
